# Effects of Different Pedaling Positions on Muscle Usage and Energy Expenditure in Amateur Cyclists

**DOI:** 10.3390/ijerph191912046

**Published:** 2022-09-23

**Authors:** Chun-Kai Tang, Ching Huang, Kai-Cheng Liang, Yu-Jung Cheng, Yueh-Ling Hsieh, Yi-Fen Shih, Hsiu-Chen Lin

**Affiliations:** 1Department of Physical Therapy and Assistive Technology, National Yang Ming Chiao Tung University, Taipei 112304, Taiwan; 2Department of Physical Therapy, Tao Yuan General Hospital, Ministry of Health and Welfare, Taoyuan 33004, Taiwan; 3Department of Physical Therapy and Graduate Institute of Rehabilitation Science, China Medical University, Taichung 406040, Taiwan

**Keywords:** muscle usage, energy expenditure, cycling, pedaling position, bike fitting

## Abstract

Background: Inappropriate cycling positions may affect muscle usage strategy and raise the level of fatigue or risk of sport injury. Dynamic bike fitting is a growing trend meant to help cyclists select proper bikes and adjust them to fit their ergometry. The purpose of this study is to investigate how the “knee forward of foot” (KFOF) distance, an important dynamic bike fitting variable, influences the muscle activation, muscle usage strategy, and rate of energy expenditure during cycling. Methods: Six amateur cyclists were recruited to perform the short-distance ride test (SRT) and the graded exercise tests (GXT) with pedaling positions at four different KFOF distances (+20, 0, −20, and −40 mm). The surface electromyographic (EMG) and portable energy metabolism systems were used to monitor the muscle activation and energy expenditure. The outcome measures included the EMG root-mean-square (RMS) amplitudes of eight muscles in the lower extremity during the SRT, the regression line of the changes in the EMG RMS amplitude and median frequency (MF), and the heart rate and oxygen consumption during the GXT. Results: Our results revealed significant differences in the muscle activation of vastus lateralis, vastus medialis, and semitendinosus among four different pedaling positions during the SRT. During GXT, no statistically significant differences in muscle usage strategy and energy expenditure were found among different KFOF. However, most cyclists had the highest rate of energy expenditure with either KFOF at −40 mm or 20 mm. Conclusions: The KFOF distance altered muscle activation in the SRT; however, no significant influence on the muscle usage strategy was found in the GXT. A higher rate of energy expenditure in the extreme pedaling positions of KFOF was observed in most amateur cyclists, so professional assistance for proper bike fitting was recommended.

## 1. Introduction

Cycling exercise has become popular in recent years. Although originally considered as a means of transportation, biking has now become a recreational activity and a type of competitive athletics. Many people start riding and become professional or amateur cyclists. Due to the long distance of biking, many cyclists suffer from various sport injuries, such as hip pain, anterior knee pain, patellar tendinitis, iliotibial band friction syndrome, quadriceps tendinitis, Achilles tendon tendinitis, etc. [1,2]. These injuries may be caused by the long periods of cycling, or by the overuse of the muscles, especially when inefficient pedaling positions are used. Therefore, it is important to recognize proper pedaling positions and further increase exercise performance.

Efficiency, as a key factor in cycling performance, is defined as the ratio of power output to energy expenditure [3]. A crank cycle can be divided into two halves, with the first half as the accelerating phase, from the highest pedal position (0°, top dead center, TDC) to the lowest (180°, bottom dead center, BDC), and the second as the recovery phase back to TDC [4]. In a crank cycle, the efficiency is associated with different pedaling directions at different cycling positions that generate forces effectively and thus crank torques, which may lead to a change of muscle activation and the level of muscle fatigue [5,6,7]. Consistent and sufficient energy supply is necessary to maintain effective muscle contraction and force generation during pedaling. Energy expenditure as determined by analyses of oxygen consumption is an important parameter in the evaluation of the exercise efficiency and the aerobic or anaerobic state [8]. The capability in aerobic metabolism of the cyclist is critical for the long period of cycling and has a strong relationship with neuromuscular fatigue [9]. It can be assumed that a proper pedaling position may facilitate an efficient generation of the crank torque that would reduce muscle activation in the short-term, and thus influence the level of muscle fatigue and the energy expenditure rate in the long run.

Bike fitting, either static or dynamic, is a good way to assess cycling posture. It is defined as ‘the detailed process of evaluating the physical and performance requirements of the cyclist, and systematically adjusting the bicycle to meet the goals and needs’ [10]. Bike fitters can use the fitting results to choose the appropriate size of the bike and adjust its components (e.g., saddle height, saddle setback). After professional bike fitting, cyclists could reduce their level of fatigue and improve exercise performance [1]. The process of static fitting includes anthropometry and bike angle measurements [11], such as the leg length, the height of the inseam, saddle height, saddle setback, seat tube angle, etc. Previous studies indicated that different static fitting parameters could change the level of discomfort, injury rate, muscle activation, muscle fatigue, and metabolic energy expenditure [12,13,14,15,16,17]. A knee flexion angle of 25 to 35° at the BDC is considered a crucial parameter for performance and injury prevention [10]. The development of dynamic bike fitting, which uses a motion capture system to record the cyclist’s posture and joint positions during the full crank cycle, enables a better understanding of the real cycling conditions than static fitting. The Retül system is a commonly used commercial dynamic bike fitting system. The parameter ‘knee forward of foot’ (KFOF) is defined as the vertical projection distance of the knee joint to the fifth metatarsal head at 90° of the pedaling cycle. It could influence the direction of the pedaling force and thus the crank torque in the accelerating phase, and then might also affect the energy efficiency and muscle fatigue [5,13]. However, there is no general consensus regarding this parameter in the bike fitting industry, and the proper value for it is still debated.

Most previous studies changed the dimensions of the bike in static fitting condition to investigate their effects on muscle activation, muscle fatigue level, and oxygen consumption. However, these changes would alter the positions of multiple body segments simultaneously. The special bike of a dynamic fitting system makes it possible to change the cyclist’s KFOF distance only while maintaining the same positions in other joints. The purposes of this study were (1) to investigate the effects of different KFOF distances on muscle activation of eight muscles in the lower limb in a short-distance ride; and (2) to investigate the effects of different KFOF distances on the muscle usage strategy and energy expenditure rate during a long period of cycling. Our hypotheses were that cycling in extreme KFOF distances that were away from the vertical line would result in less efficient pedaling, showing in higher muscle activation, easier muscle fatigue, and more energy expenditure.

## 2. Materials and Methods

### 2.1. Participants

This is an exploratory study. Male amateur cyclists were screened and recruited in this study. The inclusion criteria were: (i) age from 20 to 50, (ii) capable of completing the Taiwan Wu-Ling highest point (3275 m) challenge (altitude change: 2800 m, distance: 53 km) in 4 h, (iii) completion of the ACSM Health/Fitness Facility Pre-participation Screening Questionnaire with fewer than three items checked. This study received ethical approval from the Institutional Review Board of China Medical University Hospital (IRB number CMUH105-REC3-053). Subjects fully understood the testing procedure and signed an informed consent before the study began.

### 2.2. Instrumentations

We used a special bike (Retül Müve SL Dynamic Fit Bike) and a dynamic bike fitting system (Retül bike fitting system, Boulder, CO, USA) to control the standard cycling position while allowing us to change the KFOF parameter only. The Delsys Trigno Wireless System (Delsys Inc., Natick, MA, USA) was used to record muscle activation in the lower limb. The surface electrodes had four silver bar contacts (5 × 1 mm) spaced 10 mm apart in a rectangular configuration. Before the riding tests, the electrodes were placed at the vastus lateralis (VL), vastus medialis (VM), semimembranosus (SEMI), biceps femoris (BF), gluteus maximus (GM), tensor fascia lata (TFL), medial gastrocnemius (MG), and tibialis anterior (TA) of the right leg according to the method described in the previous study [18]. A portable metabolic system (COSMED K4b2) and a chest strap of the heart rate monitor (POLAR) were used to collect data on oxygen consumption (VO_2_) and heart rate during the cycling exercise testing.

### 2.3. Pedaling Positions

Four different pedaling positions were investigated in our study. The KFOF distance was measured at the forward pedaling position of 90° during dynamic cycling, and set at +20, 0, −20, and −40 mm as the pedaling positions in our study (the negative values indicate that the knee is behind the vertical line over the pedal spindle) (Figure 1).

### 2.4. Procedures

The experiment was divided into a short-distance ride test (SRT) and a graded exercise test (GXT). The SRT was designed to examine muscle activations in four different KFOF positions under the same riding powers. Four riding powers (120, 150, 180, 210 W) were tested on four non-consecutive days, and one power level was assigned randomly for each day. After the EMG electrodes were properly placed, the maximal voluntary isometric test (MVIC) was conducted first, using standard manual muscle testing methods, and the EMG data were recorded and analyzed for normalization. Then, the cyclist started with a warm-up phase for 5 min at 100 W in each short-distance ride. The sequence of the four pedaling positions during the SRT was randomly assigned. During the test, the cyclist was asked to maintain cadence at 90 rpm. When they reached the selected power level at each pedaling position, the myoelectric data would be recorded for 1 min. When adjusting the pedal position, the cyclist would keep pedaling at 100 W power level to prevent cooling down.

After a 5-min rest, the metabolic system (COSMED K4b2) and the POLAR heart rate monitor were equipped. Then, the GXT was conducted to examine the muscle activation level (%MVC), muscle fatigue (MF change), and energy expenditure during a long-distance ride. One randomly assigned pedaling position was tested on each day. After proper dynamic fitting, the riding power for the GXT was set starting at 120 W and increased incrementally by 10 W every minute until the cyclist could no longer continue. During the test, each cyclist was asked to maintain a ride cadence at around 90 rpm. If the cadence was less than 80 rpm, the researchers would remind and encourage the cyclist. However, if the cadence dropped to less than 70 rpm, the test would be ended. In the cool down phase, the cyclist would continue riding 120 W until the heart rate returned to the baseline at the start of the GXT. The main reason to stop the test—i.e., due to leg soreness or breathlessness—would be documented.

### 2.5. Data Reduction and Analysis

The recorded myoelectric signals in the middle 30 s of every testing minute in the SRT and GXT were used for further analysis. The oxygen consumption (VO_2_) was calculated and recorded breath-by-breath by the metabolic system, and the heart rate measured by the POLAR strap was recorded every 30 s during the whole GXT.

The pedaling cycle was first identified using the thigh acceleration and gyroscopic data collected simultaneously by the Trigno smart sensor on the VL. The raw EMG data were first analyzed with the bandpass filter (10 to 500 Hz), and then the root-mean-square (RMS) values were calculated by using a moving window of 0.02 s with 0.01 s overlapped. These data were then normalized with the MVIC value of each muscle after the same signal processing to represent the amplitude of the muscle activation. For every 30 s data of every testing minute, the peaks of the normalized RMS extracted from the pedaling cycles were then averaged.

For the GXT, the filtered EMG data were also processed to obtain the median frequency (MF) of each muscle—reflecting the shift in recruitment of muscle fiber types—in each pedaling cycle and then averaged in every minute. Since we set the increment of the cycling power constant, the changes in these variables were expected to have a linear trend. Therefore, the mean RMS, MF, VO_2_, and heart rate of each testing minute during the GXT was further analyzed using linear regression, and the coefficient of the variable in the regression equation represented their changing trend as the riding power and time increased (Figure 2). SPSS software (IBM SPSS Inc., Chicago, USA) was used for statistical analysis with the significant level set at 0.05. With the limited sample size, the Friedman non-parametric test was used to compare the differences in muscle activation levels in the SDT among four different pedaling powers and positions and the changing trends in RMS, MF, VO_2_, and heart rate during the GXT among four different pedaling positions. The post hoc test was conducted by the Wilcoxon sign rank test.

## 3. Results

Six male cyclists (age 32.5 ± 2.7 years) participated voluntarily and completed the tests in this study. Their mean height and weight were 173.7 ± 1.5 cm and 72.0 ± 11.0 kg, respectively. During the SRT, we found significant differences in VL and VM among four different pedaling positions at 180 W workload and SM at 210 W workload. For VL and VM muscles, the activation during cycling with KFOF at 20 mm was the greatest, while for the SEMI muscle, the activation with KFOF at 0 mm was the least. Comparing among the four increasing workloads, significantly increasing trends in VM muscle activation at KFOF at +20, 0, and −40 mm were found. The muscle activation of the VM would increase significantly as the workload increased from 120 to 210 W at KFOF at +20 and 0 mm and in comparison of 210 W with 150 W at KFOF at −40 mm (Table 1).

During the GXT, no statistical significance was found when comparing the coefficients of the regression line of the normalized mean RMS of the eight muscles among the four pedaling positions (Table 2). A similar finding was observed for the regression line of the MF, except for the TA. There was a significantly lower coefficient in the regression line of the MF of the TA in the pedaling position with KFOF at 0 mm compared to the others (Table 2). In the energy expenditure, there were no significant differences in the coefficients in the regression line of heart rate and VO_2_ among the four pedaling positions (Table 2).

With the small sample size in this study, the following statement would focus on some interesting features during the GXT. Regarding the muscle usage, we chose the VL, VM, and SEMI muscles, which showed statistical significance among different pedaling positions or powers in SRT, for further exploration. In Figure 3a–c, we observe that all the coefficients of the regression lines for the RMS amplitude in VM, VL, and SEMI in subject four were greater than 0 and the coefficients of the MF (Figure 3d–f) were smaller than 0, showing that he adopted a same muscle usage strategy in all different pedaling positions. Moreover, he reported a same reason of muscle soreness for the termination of the GXT. In contrast, subject six had all coefficients of the regression lines greater than 0 for the MF of the three muscles (Figure 3d–f) and reported shortness of breath as the reason for terminating the GXT in all pedaling positions. Other cyclists showed no such consistent strategy in all four different pedaling positions.

Some characteristics of the coefficient in the regression line of the heart rate and VO_2_ during the GXT were observed. For the heart rate, subjects two and three had the greatest coefficient with KFOF at −40 mm and subject four with KFOF at +20 mm, which represented having the fastest increasing heart rate at a certain position as the workload rises in each subject (Figure 4a). Similarly, for the VO_2_, subjects one, two, and four had the greatest VO_2_ coefficient with KFOF at −40 mm, and subject five with KFOF at 20 mm (Figure 4b)

## 4. Discussion

This study aimed to investigate how different KFOF distances could influence muscle activation during the SRT, and the muscle usage strategy and energy expenditure during the GXT. In the SRT, we found the VL and VM had lesser muscle activation as the KFOF distance decreased—that is, the knee joint was more posterior to the 5th metatarsal when the power level was set at 180 W (Table 1). The pedaling torque was associated with the crank length and effective force, which was defined as the vertical force of the total pedaling force [19]. As the knee joint moved posteriorly to the 5th metatarsal, the knee flexion angle would be decreased, and the length of the VL and VM muscles would become slightly shorter. At 90° of the pedaling cycle, the knee flexion angle is around 63 degrees, which is located near the plateau of the optimal length for VL and VM [20] to generate the same muscle force with less muscle activation. We also found that only the VM activation would increase as the power level rose. VM plays an important role in controlling the patellofemoral gliding [1,2]. The result indicates that the VM muscle is a key muscle for pedaling capability and the stabilization of the knee joint complex during the acceleration phase. In contrast, the smallest SEMI muscle activation was found with KFOF at 0 mm during a power level at 210 W. It may not function for the power generation, but, as a bi-articular muscle, it helps the pedaling cycle to proceed in either the acceleration or recovery phase. However, over-activation of the SEMI muscle during cycling would result in pelvic posterior tilt and increased muscle tension in the lumbar region and might cause lower back pain during long-distance cycling. Therefore, placing this muscle around its efficient position could avoid over-activation and fatigue.

The cyclists terminated the GXT when the power reached around 310 W, much higher than that in the SRT. We analyzed the shift of the MF to show the possible muscle fatigue and the changing trends of the RMS value to detect different muscle usage strategies. However, the results showed no significant differences in either the MF or RMS among four pedaling positions in all major muscles during cycling, except for the TA (Table 2). A significantly smaller coefficient in the regression line of MF in the TA was found with KFOF at 0 mm. Although the ankle joint is not considered a major power-generating joint, it is still significant to the force transmission. A previous study found that the TA muscle tends to activate prior to the top dead center during cycling [21] and functions as the energy transformer to the pedal with quadricep muscles [22]. The reasons the major force-generating muscles did not reveal any significant tendency toward the shift of the MF may be because of the increasing power level during the GXT, which required more powerful muscle fiber recruitment (i.e., type II muscle fibers), resulting in a higher MF. Previous studies found that when the fatigue test was conducted at the same power level, the MF would significantly decrease; however, there was no significant change in the GXT [14,23].

When investigating the muscle usage strategy in each subject, some features were observed (Figure 3). First, an increased coefficient of RMS value as the GXT progressed indicates the subject tended to recruit more new motor units, which results in incremental increases of RMS values [24], like subject four. At the termination of the GXT, he reported the same reason of muscle soreness, suggesting that the major limitation came from the muscle. We found that subjects who terminated the GXT due to muscle soreness in the legs tended to have a decreasing trend (negative coefficient of the regression line) of MF, and those who reported the reason of shortness of breath tended to have an increasing trend (positive coefficient) as the power increased. Subject six reported the same reason of shortness of breath for terminating the GXT in all pedaling positions, and all the coefficients were positive for the MF and various patterns in muscle activation. This suggested that the limitation for subject six came from the cardiopulmonary endurance rather than the muscle function. These phenomena conformed with the large variability in the shift of the MF and thus explain why we could not detect any statistical difference in different pedaling positions. Regarding energy expenditure, we observed the highest coefficients of the regression line in VO_2_ and/or heart rate with KFOF at 20 and −40 mm in most of our subjects during the GXT (Figure 4). The increased rate in oxygen consumption may lead cyclists to reach the anaerobic threshold faster and increase the heart rate quicker [23]. Heart rate is a convenient indicator of energy expenditure that cyclists commonly measure during long periods of cycling.

To the best of our knowledge, no other study has used KFOF as a parameter to assess its effects on cycling efficiency. Previous studies commonly changed the seat tube angle, saddle height, or saddle setback position to observe their effects on the level of muscle activation, muscle fatigue, and energy expenditure [12,14,15,16,23]. However, the relationships among pedaling positions, muscle fatigue, and energy consumption are still uncertain. Our study showed results consistent with most studies, according to which extreme pedaling positions away from the vertical line, such as KFOF at +20 mm or −40 mm, would be prone to resulting in muscle fatigue or increased muscle usage and increased energy expenditure.

The current study demonstrated that the KFOF parameter could alter muscle activations and can be considered a good parameter during dynamic bike fitting. The first limitation of this study was the small sample size. These cyclists demonstrated various responses in muscle usage strategy and energy expenditure in GXT to different KFOF distances; however, some interesting features were observed. After retrospective power analysis, we suggested a larger sample size of 16 cyclists may be needed to show the effects of different pedaling positions. Secondly, we recruited amateur cyclists instead of professional athletes. These amateur cyclists do not receive professional training and are not in sports-related occupations, so they tend to have their own preferential pedaling strategies and physiological limitations regarding the increasing power level in the GXT. Thirdly, the shift of the MF in the muscle activation may not be appropriate for assessing muscle fatigue during the GXT. Future research could recruit more well-trained cyclists to investigate the effects in different pedaling positions for the best cycling efficiency.

## 5. Conclusions

In conclusion, “knee forward of foot” (KFOF) distance could alter the muscle activation of VL, VM, and SEMI in the SRT. Only the activation level in VM was found to be increased with the increasing workloads in the SRT. With various responses in the amateur cyclists in the GXT, no significant alteration or consistent pattern in the muscle usage strategy was found. However, a higher rate of energy expenditure in extreme pedaling positions of KFOF at +20 mm or −40 mm was observed in the GXT for most cyclists. It is recommended that cyclists find professional assistance for the proper bike fitting.

## Figures and Tables

**Figure 1 ijerph-19-12046-f001:**
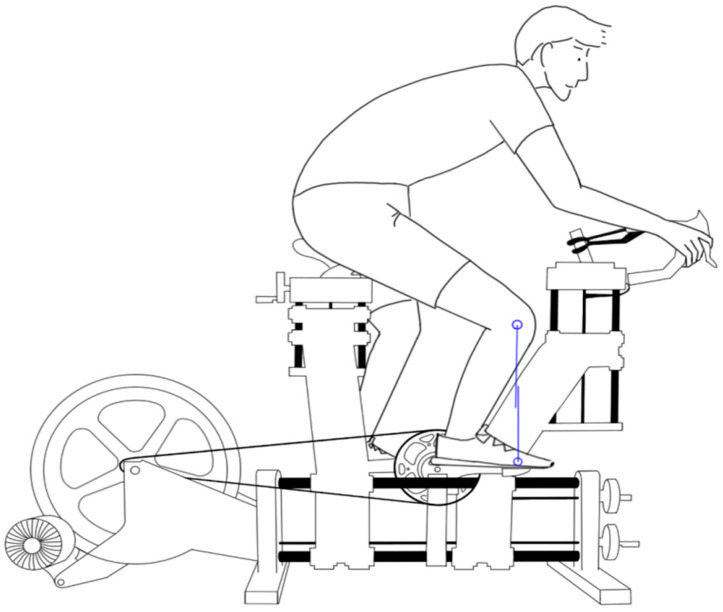
The dynamic fitting bike and system was used to adjust the cyclist’s pedaling position. The knee-forward-of-foot (KFOF) distance was calculated by the vertical lines of the knee and the foot markers (blue lines) during dynamic cycling.

**Figure 2 ijerph-19-12046-f002:**
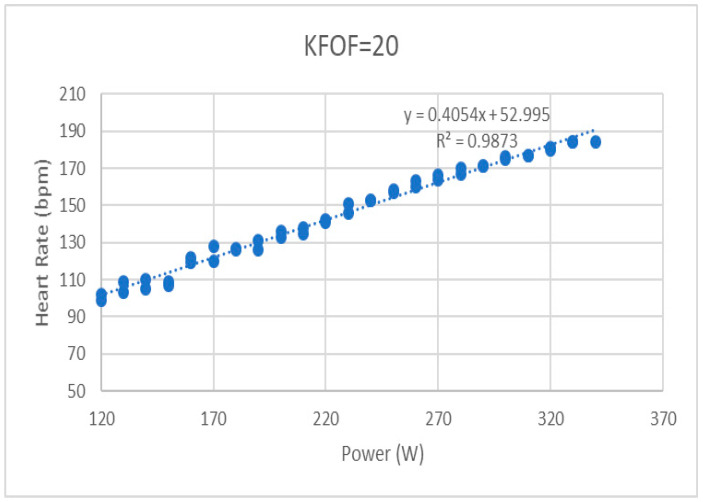
An example of a regression line of the heart rate with KFOF at 20 mm position.

**Figure 3 ijerph-19-12046-f003:**
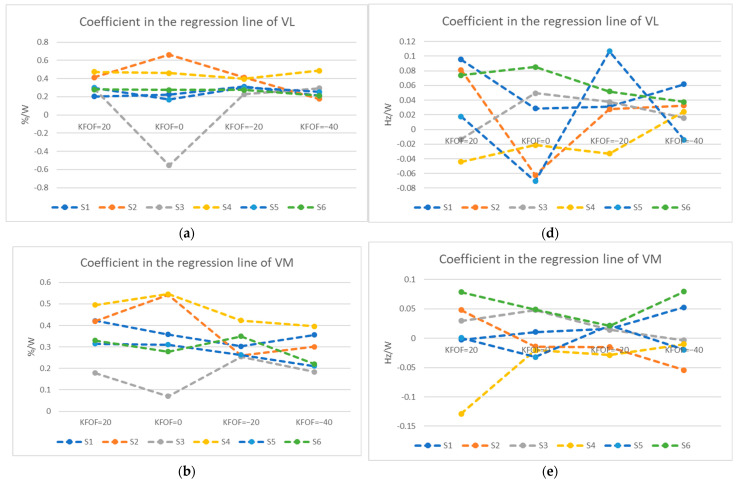
(**Left**) Coefficient in the regression line of the RMS value in GXT for (**a**) Vastus lateralis (VL); (**b**) Vastus medialis (VM); (**c**) semitendinosus (SEMI); (**Right**) Coefficient in the regression line of median frequency in GXT for (**d**) Vastus lateralis (VL); (**e**) Vastus medialis (VM); (**f**) semitendinosus (SEMI).

**Figure 4 ijerph-19-12046-f004:**
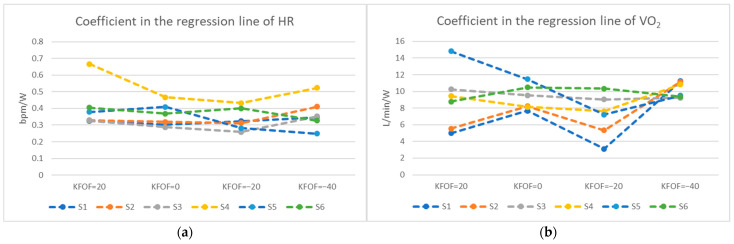
Coefficient in the regression line of (**a**) heart rate (HR); (**b**) oxygen consumption (VO_2_).

**Table 1 ijerph-19-12046-t001:** The normalized RMS value (%MVC, mean ± SD) during different pedaling postures and workloads in the short-distance ride test (SRT).

Muscles	Pedaling Position	120 W	150 W	180 W	210 W
Vastus lateralis (VL)	KFOF = 20	76.46 ± 36.65	114.93 ± 119.59	80.88 ± 10.47 *^,a,b^	81.12 ± 6.96
	KFOF = 0	77.88 ± 37.94	129.55 ± 145.34	74.49 ± 5.81 *^,a^	80.82 ± 11.28
	KFOF = −20	71.82 ± 22.25	120.27 ± 117.40	72.58 ± 6.18 *^,d^	79.87 ± 7.93
	KFOF = −40	88.80 ± 36.84	119.88 ± 124.90	66.64 ± 9.19 *^,b,d^	79.04 ± 17.33
Vastus medialis (VM)	KFOF = 20	61.61 ± 12.34 ^†,e^	68.15 ± 28.40 ^†,f,g^	77.60 ± 9.09 *^,b,†,f^	91.43 ± 13.00 ^†,e,g^
	KFOF = 0	60.47 ± 12.03 ^†,e^	65.65 ± 25.10 ^†,f,g^	74.23 ± 8.45 *^,c,†,f^	90.52 ± 18.93 ^†,e,g^
	KFOF = −20	61.28 ± 9.54	65.71 ± 23.28	71.85 ± 7.66 *^,d^	87.77 ± 11.96
	KFOF = −40	64.34 ± 13.12	65.36 ± 29.62 ^†,g^	68.51 ± 6.02 *^,c,d^	86.36 ± 10.95 ^†,g^
Semitendinosus (SEMI)	KFOF = 20	42.57 ± 19.96	21.03 ± 13.51	20.08 ± 13.89	20.04 ± 9.08 *^,a^
	KFOF = 0	44.13 ± 23.65	20.50 ± 14.20	21.63 ± 15.83	18.75 ± 7.57 *^,ac^
	KFOF = −20	42.72 ± 16.54	22.71 ± 12.44	18.16 ± 10.65	21.54 ± 8.80 *
	KFOF = −40	44.96 ± 19.96	21.89 ± 13.21	18.40 ± 10.07	19.81 ± 7.65 *^,c^
Biceps femoris(BF)	KFOF = 20	48.20 ± 16.92	69.48 ± 48.30	48.84 ± 25.89	57.75 ± 40.96
	KFOF = 0	48.23 ± 18.68	66.06 ± 49.27	45.73 ± 25.73	59.13 ± 40.53
	KFOF = −20	47.29 ± 21.74	64.77 ± 46.41	41.17 ± 25.62	57.28 ± 39.67
	KFOF = −40	50.22 ± 14.95	26.70 ± 67.53	50.22 ± 14.95	26.70 ± 67.53
Tensor fascia lata (TFL)	KFOF = 20	71.83 ± 75.74	38.67 ± 19.50	215.97 ± 265.80	34.23 ± 16.52
	KFOF = 0	71.30 ± 71.97	38.92 ± 19.78	129.23 ± 142.97	31.59 ± 15.89
	KFOF = −20	112.40 ± 139.09	40.40 ± 22.59	229.77 ± 255.15	44.52 ± 36.08
	KFOF = −40	48.05 ± 44.12	9.57 ± 107.94	273.61 ± 343.88	39.94 ± 24.80
Gluteus maximus (GM)	KFOF = 20	85.69 ± 44.64	99.19 ± 60.12	87.43 ± 26.36	107.49 ± 16.44
	KFOF = 0	68.03 ± 32.08	105.83 ± 50.06	97.37 ± 8.69	115.64 ± 8.46
	KFOF = −20	80.60 ± 30.88	114.61 ± 39.30	92.77 ± 9.95	134.03 ± 34.88
	KFOF = −40	86.68 ± 35.86	89.33 ± 22.90	93.29 ± 23.12	131.95 ± 43.62
Medial gastrocnemius (MG)	KFOF = 20	107.42 ± 27.85	88.44 ± 19.11	94.83 ± 18.40	81.23 ± 27.19
	KFOF = 0	88.91 ± 24.31	85.99 ± 17.97	98.81 ± 19.97	70.75 ± 28.64
	KFOF = −20	92.68 ± 21.89	96.75 ± 28.77	93.69 ± 14.16	84.12 ± 41.84
	KFOF = −40	133.08 ± 48.16	92.84 ± 28.93	94.75 ± 26.21	77.73 ± 28.89
Tibialis anterior (TA)	KFOF = 20	91.15 ± 62.82	18.81 ± 5.33	174.98 ± 359.42	73.69 ± 86.47
	KFOF = 0	108.33 ± 80.84	15.25 ± 6.33	139.42 ± 253.85	88.90 ± 112.23
	KFOF = −20	49.39 ± 13.48	17.34 ± 3.48	50.57 ± 57.17	106.98 ± 125.48
	KFOF = −40	89.28 ± 61.95	13.76 ± 6.31	64.23 ± 81.73	114.21 ± 104.65

* Significant difference (*p* < 0.05) among four pedaling positions using Friedman non-parametric test. ^†^ Significant difference (*p* < 0.05) among four different power levels using Friedman non-parametric test. ^a^ Significant difference (*p* < 0.05) between KFOF = 20 vs. KFOF = 0 using post hoc test. ^b^ Significant difference (*p* < 0.05) between KFOF = 20 vs. KFOF = −40 using post hoc test. ^c^ Significant difference (*p* < 0.05) between KFOF = 0 vs. KFOF = −40 using post hoc test. ^d^ Significant difference (*p* < 0.05) between KFOF = −20 vs. KFOF = −40 using post-hoc test. ^e^ Significant difference (*p* < 0.05) between 120 W vs. 210 W using post hoc test. ^f^ Significant difference (*p* < 0.05) between 150 W vs. 180 W using post hoc test. ^g^ Significant difference (*p* < 0.05) between 150 W vs. 210 W using post hoc test.

**Table 2 ijerph-19-12046-t002:** The coefficients (mean ± SD) in the regression lines of the normalized root-mean-square (RMS) amplitude (%MVC) and median frequency (MF) of the eight muscles, heart rate, and oxygen consumption (VO_2_) during the graded exercise test (GXT).

		KFOF = 20	KFOF = 0	KFOF = −20	KFOF = −40
Vastus lateralis	RMS	0.33 ± 0.10	0.21 ± 0.41	0.32 ± 0.07	0.27 ± 0.11
(VL)	MF	0.04 ± 0.06	0.14 ± 0.39	0.04 ± 0.05	0.03 ± 0.03
Vastus medialis	RMS	0.36 ± 0.11	0.35 ± 0.18	0.31 ± 0.07	0.28 ± 0.09
(VM)	MF	0.00 ± 0.07	0.14 ± 0.34	0.01 ± 0.02	0.01 ± 0.05
Semitendinosus	RMS	0.23 ± 0.12	0.14 ± 0.16	0.16 ± 0.10	0.45 ± 0.52
(SEMI)	MF	0.01 ± 0.07	0.12 ± 0.28	0.01 ± 0.07	−0.01 ± 0.07
Biceps femoris	RMS	0.29 ± 0.09	0.25 ± 0.10	0.28 ± 0.09	0.26 ± 0.10
(BF)	MF	0.05 ± 0.02	0.15 ± 0.32	0.06 ± 0.07	0.07 ± 0.07
Tensor fascia lata	RMS	0.21 ± 0.40	0.26 ± 0.15	0.18 ± 0.17	0.09 ± 0.27
(TFL)	MF	0.02 ± 0.10	0.15 ± 0.35	0.06 ± 0.06	0.05 ± 0.16
Gluteus maximus	RMS	0.25 ± 0.13	0.21 ± 0.38	0.21 ± 0.14	0.28 ± 0.17
(GM)	MF	0.09 ± 0.12	0.18 ± 0.37	0.10 ± 0.10	0.04 ± 0.05
Medial	RMS	0.08 ± 0.17	−0.04 ± 0.17	0.08 ± 0.26	−0.04 ± 0.05
gastrocnemius (MG)	MF	−0.02 ± 0.05	0.04 ± 0.22	0.02 ± 0.15	0.04 ± 0.06
Tibialis anterior	RMS	0.08 ± 0.50	−0.00 ± 0.29	−0.41 ± 0.69	−0.01 ± 0.41
(TA)	MF *	0.05 ± 0.08 ^a^	−0.10 ± 0.10 ^a,b^	0.03 ± 0.13	0.02 ± 0.05 ^b^
Heart rate		0.36 ± 0.07	0.41 ± 0.13	0.34 ± 0.07	0.37 ± 0.09
VO_2_		9.25 ± 1.50	8.97 ± 3.57	7.10 ± 2.59	10.21 ± 0.94

* Significant difference (*p* < 0.05) between four pedaling positions using Friedman non-parametric test. ^a^ Significant difference (*p* < 0.05) between KFOF = 20 vs. KFOF = 0 using post-hoc test. ^b^ Significant difference (*p* < 0.05) between KFOF = 0 vs. KFOF = −40 using post-hoc test.

## Data Availability

The data presented in this study are available on request from the corresponding author.

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
