# Peer review of "Effects of Different Pedaling Positions on Muscle Usage and Energy Expenditure in Amateur Cyclists"

_ijerph, 2022, doi:10.3390/ijerph191912046_

Round 1

Reviewer 1 Report

Overall, the manuscript is considered correct and interesting.

However, it would have been interesting to provide more information about the participants: how were they recruited? Did they receive any kind of compensation? As well as providing information on some sociodemographic variables (what they do for example, if any of them do something related to sports).

Author Response

They were recruited through the recruitment poster posted in the bike shop and did not receive any kinds of compensation. We didn’t collect their sociodemographic variables, however, their occupations didn’t relate to sport and possibly influence the results of our study. We have added the information into the last paragraph of the Discussion. Thank you very much.

Reviewer 2 Report

This study is identified as an exploratory study to examine the muscle activation pattern changes in lower limb muscles during a graded exercise test performed at 4 different pedal positions.

General comment- Since this is an exploratory study, statistical power in results is difficult to attain, especially with EMG data with very high standard deviations. However if that is what you use, then one cannot stretch beyond those results to talk about "increasing trends" or other comments suggesting outcomes that have not been born out in the statistical analysis. Please focus the results and discussion to be more succinct. Also, consider how much of the individual results are providing insights. The wide range of individual results looks more like noise, and not various strategies of recruitment.

Abstract

Line 24- I think you mean heart rate.... not heart

The term "muscle usage" is unclear. Muscle usage strategy makes more sense. I am reading this term as meaning the pattern of sequential use of muscles in the cycling stroke. However it is a bit unclear exactly what you mean by muscle usage. Please clarify

The results indicated in the abstract do not relate to the conclusion:

- results- Results show no significant difference in muscle usage.....

-conclusion- KFOF could alter muscle usage.........

Introduction

P. 2 lines 59-62 ... "proper pedaling position may facilitate efficient generation of crank torque which would change the muscle usage.." What do you mean by this? Do you mean a more efficient pedal crank will reduce muscle activation? Again, the use of the term "muscle usage" is confusing, and "change" is a non descriptive term.

Methods

P. 4 line 148 - GXT conducted to examine the muscle usage".. Unclear how the testis doing this. Does muscle usage mean fatigue (MF change), does it mean muscle activation (RMS% MVC) or does it mean the order of activation within the cycle stroke?

Line 172- median frequency- to represent "tendency in using fibers" Please clarify this. MF can be used to examine fatigue, but also a shift in fiber recruitment. However during a GXT you may have BOTH fatigue and changes in fiber recruitment due to increasing load. Please clarify purpose of the MF measurement

Results

The results are difficult to follow, as it contains measures of statistical mean data, and also individual data. Can the results be made more succinct to only the significant results or most interesting within individual results? Otherwise much of it looks like noise in the data.

Table 1 line 194 The statement "significantly increasing trends" Unclear. Is this based on an ANOVA? Is the trend across workrates within the Vastus Medialis? or is it just within the 180W workrate, since that is the only area where the table is suggesting significant differences. Please clarify this statement and be sure to only indicate changes that are born out of statistical significance.

Figures 3: a-f- These various graphs of individual data really look like noise, and it is difficult to not anything that would suggest differences in subject responses. Do you have more descriptive data about the subjects that could suggest why one might see differences. Example, height, crural ratio, limb lengths.. etc

p. 9 line 303- unclear- did you run an ANOVA to determine that VM activation increased with power output? This was not clear looking at Table 1

Discussion-

Please shorten the discussion to the significant statistical results you identified and then any particular interesting individual result that may provide insight for a follow-up study. If this study is an exploratory study, then the reader will gain insight as to what might be of use to investigate in a larger study. Currently the discussion is too long and wide ranging, and it is difficult to draw any insight from it. Conclusions need to be consistent with found results and not be speculative.

Author Response

Response to Reviewer 2 Comments

 Comments and Suggestions for Authors

This study is identified as an exploratory study to examine the muscle activation pattern changes in lower limb muscles during a graded exercise test performed at 4 different pedal positions.

Point 1: General comment- Since this is an exploratory study, statistical power in results is difficult to attain, especially with EMG data with very high standard deviations. However if that is what you use, then one cannot stretch beyond those results to talk about "increasing trends" or other comments suggesting outcomes that have not been born out in the statistical analysis. Please focus the results and discussion to be more succinct. Also, consider how much of the individual results are providing insights. The wide range of individual results looks more like noise, and not various strategies of recruitment.

Response 1: Thank you very much for the valuable opinions. We have revised the results and discussion and focus only on some interesting features.

Point 2: Abstract  Line 24- I think you mean heart rate.... not heart

Response 2: Revised. Thank you.

Point 3: The term "muscle usage" is unclear. Muscle usage strategy makes more sense. I am reading this term as meaning the pattern of sequential use of muscles in the cycling stroke. However it is a bit unclear exactly what you mean by muscle usage. Please clarify

Response 3: Thank you for the suggestion. We used the muscle usage “pattern” to include mean muscle activation (RMS% MVC) and the change of the median frequency. We agreed that muscle usage “strategy” would be more appropriate. The manuscript has been revised accordingly.

Point 4: The results indicated in the abstract do not relate to the conclusion:

- results- Results show no significant difference in muscle usage.....

-conclusion- KFOF could alter muscle usage.........

Response 4: We had revised the Conclusion.

Point 5: P. 2 lines 59-62 ... "proper pedaling position may facilitate efficient generation of crank torque which would change the muscle usage.." What do you mean by this? Do you mean a more efficient pedal crank will reduce muscle activation? Again, the use of the term "muscle usage" is confusing, and "change" is a non descriptive term.

Response 5: Yes, we were trying to connect the proper pedaling position to a more efficient use of muscle and thus to reduce muscle fatigue and energy expenditure in the long run. We have changed the sentence as suggestion.

Point 6: P. 4 line 148 - GXT conducted to examine the muscle usage".. Unclear how the test is doing this. Does muscle usage mean fatigue (MF change), does it mean muscle activation (RMS% MVC) or does it mean the order of activation within the cycle stroke?

Response 6: We examined muscle activation level (%MVC) and muscle fatigue (MF change). The manuscript has been revised.

Point 7: Line 172- median frequency- to represent "tendency in using fibers" Please clarify this. MF can be used to examine fatigue, but also a shift in fiber recruitment. However during a GXT you may have BOTH fatigue and changes in fiber recruitment due to increasing load. Please clarify purpose of the MF measurement

Response 7: We would like to measure the level of muscle fatigue and also demonstrated the shift in fiber recruitment.

Point 8: The results are difficult to follow, as it contains measures of statistical mean data, and also individual data. Can the results be made more succinct to only the significant results or most interesting within individual results? Otherwise much of it looks like noise in the data.

Response 8: We had revised the results by deleting most of the individual observation. Only some interesting features were presented.

Point 9: Table 1 line 194 The statement "significantly increasing trends" Unclear. Is this based on an ANOVA? Is the trend across workrates within the Vastus Medialis? or is it just within the 180W workrate, since that is the only area where the table is suggesting significant differences. Please clarify this statement and be sure to only indicate changes that are born out of statistical significance.

Response 9: The “significantly increasing treads” were based on the Friedman non-parametric test comparing among positions and also the post-hoc results using Wilcoxon sign rank test. The result revealed that as the workload increased, the muscle activity of Vastus Medialis would also increase. We had revised the statement and added the statistical method of the post-hoc test in the method section. The notations in Table 1 have been modified, too.

Point 10: Figures 3: a-f- These various graphs of individual data really look like noise, and it is difficult to not anything that would suggest differences in subject responses. Do you have more descriptive data about the subjects that could suggest why one might see differences. Example, height, crural ratio, limb lengths.. etc

Response 10: With limited resources, we recruit the cyclists with similar body build, therefore, we didn’t collect those detail descriptive data. We have revised the results by focusing on some consistent strategy in the specific cyclists.

Point 11: p. 9 line 303- unclear- did you run an ANOVA to determine that VM activation increased with power output? This was not clear looking at Table 1

Response 11: We use the Friedman non-parametric test to determine the differences among power outputs and pedaling positions.  The post-hoc using Wilcoxon sign rank test were also conducted. The notations in Table 1 have been modified.

Point 12: Please shorten the discussion to the significant statistical results you identified and then any particular interesting individual result that may provide insight for a follow-up study. If this study is an exploratory study, then the reader will gain insight as to what might be of use to investigate in a larger study. Currently the discussion is too long and wide ranging, and it is difficult to draw any insight from it. Conclusions need to be consistent with found results and not be speculative.

Response 12: We have revised the results and discussion and focus only on some interesting features.

Round 2

Reviewer 2 Report

English language editing is needed in order to fully comprehend the explanations provided in the discussion. As it reads currently, this is quite difficult to follow.

The Median frequency results , which showed no change seem to conflict with the expected use of the MF. If changes reflect fiber recruitment, or changes reflect fatigue, and no actual changes were seen during the GXT... then this is a confusing result.

The discussion is VERY long. With an exploratory study and few subjects, can't the data shown bee reduced down to the most interesting? There is such a volume of data that any interesting finding is lost in the volume. 

Author Response

Comments and Suggestions for Authors

Point 1:

English language editing is needed in order to fully comprehend the explanations provided in the discussion. As it reads currently, this is quite difficult to follow.

Response 1: It will be sent to the Language Editing Services from MDPI.

Point 2:

The Median frequency results , which showed no change seem to conflict with the expected use of the MF. If changes reflect fiber recruitment, or changes reflect fatigue, and no actual changes were seen during the GXT... then this is a confusing result.

Response 2: We had revised the contents regarding the median frequency.

Point 3:

The discussion is VERY long. With an exploratory study and few subjects, can't the data shown be reduced down to the most interesting? There is such a volume of data that any interesting finding is lost in the volume.

Response 3: We had revised our manuscript. The word count has been reduced to 3537.
